# Tissue Acidosis Associated with Ischemic Stroke to Guide Neuroprotective Drug Delivery

**DOI:** 10.3390/biology9120460

**Published:** 2020-12-11

**Authors:** Orsolya M. Tóth, Ákos Menyhárt, Rita Frank, Dóra Hantosi, Eszter Farkas, Ferenc Bari

**Affiliations:** Department of Medical Physics and Informatics, Faculty of Medicine and Faculty of Science and Informatics, University of Szeged, H-6720 Szeged, Hungary; m.tothorsolya@gmail.com (O.M.T.); menyhartakos89@gmail.com (Á.M.); ritafrank993@gmail.com (R.F.); dora.hantosi@gmail.com (D.H.); bari.ferenc@med.u-szeged.hu (F.B.)

**Keywords:** acidosis, cerebral ischemia, penumbra, nanomedicine, neuroprotection, pH imaging, smart nanosystem, spreading depolarization, tissue pH

## Abstract

**Simple Summary:**

Ischemic stroke is caused by the blockade of a blood vessel in the brain. Consequently, the brain region supplied by the blocked vessel suffers brain damage and becomes acidic. Here we provide a summary of the causes and consequences of acid accumulation in the brain tissue. Ischemic stroke requires immediate medical attention to minimize the damage of brain tissue, and to save function. It would be desirable for the medical treatment to target the site of injury selectively, to enrich the site of ongoing injury with the protective agent, and to avoid undesirable side effects at the same time. We propose that acid accumulation at the sight of brain tissue injury can be used to delineate the region that would benefit most from medical treatment. Tiny drug carriers known as nanoparticles may be loaded with drugs that protect the brain tissue. These nanoparticles may be designed to release their drug cargo in response to an acidic environment. This would ensure that the therapeutic agent is directed selectively to the site where it is needed. Ultimately, this approach may offer a new way to treat stroke patients with the hope of more effective therapy, and better stroke outcome.

**Abstract:**

Ischemic stroke is a leading cause of death and disability worldwide. Yet, the effective therapy of focal cerebral ischemia has been an unresolved challenge. We propose here that ischemic tissue acidosis, a sensitive metabolic indicator of injury progression in cerebral ischemia, can be harnessed for the targeted delivery of neuroprotective agents. Ischemic tissue acidosis, which represents the accumulation of lactic acid in malperfused brain tissue is significantly exacerbated by the recurrence of spreading depolarizations. Deepening acidosis itself activates specific ion channels to cause neurotoxic cellular Ca^2+^ accumulation and cytotoxic edema. These processes are thought to contribute to the loss of the ischemic penumbra. The unique metabolic status of the ischemic penumbra has been exploited to identify the penumbra zone with imaging tools. Importantly, acidosis in the ischemic penumbra may also be used to guide therapeutic intervention. Agents with neuroprotective promise are suggested here to be delivered selectively to the ischemic penumbra with pH-responsive smart nanosystems. The administered nanoparticels release their cargo in acidic tissue environment, which reliably delineates sites at risk of injury. Therefore, tissue pH-targeted drug delivery is expected to enrich sites of ongoing injury with the therapeutical agent, without the risk of unfavorable off-target effects.

## 1. Introduction

Acute cerebral ischemia is a significant cause of irreversible brain injury. Global cerebral ischemia is created by the complete cessation of blood supply to the brain associated most frequently with cardiac arrest. Conversely, focal cerebral ischemia, which accounts for 80 % of all stroke cases, is caused by the obstruction of a cerebral blood vessel by atherosclerosis or embolization [1]. Most often, cerebral ischemia is incomplete. Adjacent to the core of the injury, residual blood flow persists, the local cerebral blood flow (CBF) ranging between 15–25 mL/100 g/min or 20–40% relative to baseline [2,3]. This narrow tissue band embracing the infarcted core has been known as the ischemic penumbra [3]. In contrast with the necrotic core, the penumbra consists of electrophysiologically inactive but viable and, most importantly, salvageable tissue [3,4,5,6], which places it in the center of ischemic neuroprotective therapy.

It is important to realize that the penumbra evolves dynamically in space and time [7]. Over the hours following the cerebrovascular occlusion, a significant volume of the peri-infarct penumbra may be recruited to the necrotic infarction as spreading depolarizations (SDs) [8,9] propagate across tissue at risk [7,10,11,12,13]. In fact, recurrent SDs, which are appreciated to arise at inner penumbra from minutes to days after the primary impact, have been understood as the principal mechanism of lesion progression in the acutely injured human brain [14,15]. In particular, SDs occur in hypoperfused nervous tissue due to metabolic supply demand mismatch [16], and in turn, exacerbate the ischemia-related metabolic burden [17]. The SD-linked metabolic challenge is faithfully reflected by tissue acidosis, the focus of the present review [18,19]. Importantly, the metabolic crisis associated with SD [20] may prove fatal to the penumbra tissue [21,22]. Over recent years, the opinion has been formulated that the pattern of SD recurrence should be considered as a biomarker of metabolic failure in neurointensive care [23], and SDs have been proposed as a therapeutic target in the management of acute brain injury, including ischemic stroke [24].

Even though there is a pressing need for the successful treatment of focal cerebral ischemia, clinical therapeutic options have been limited to the application of tissue plasminogen activator (tPA), or mechanical thrombectomy [25]. Yet, a high proportion of ischemic stroke patients cannot benefit from these interventions. Additional neuroprotective strategies are thus urgently required to alleviate ischemic neuronal injury when recanalization is not an option, or as supportive, adjuvant treatment after recanalization to mitigate reperfusion injury.

The development of drug delivery systems may be a key element to achieve successful therapy [26,27,28]. The particular advantages of drug delivery systems in brain injury include that (i) blood-brain barrier permeability for a specific drug is substantially increased, (ii) drug delivery can be targeted selectively to sites at risk of injury, (iii) these systems carry the potential of gradual drug release to elongate drug exposure, and (iv) local drug concentration may become high enough to exert therapeutic effect, without the risk of drug accumulation in non-target tissues, which carries the risk of undesirable side effects or off-target toxicity. “Smart” nanonsystems may meet all these requirements, and provide an effective tool in the management of ischemic stroke [29,30].

This review is dedicated to provide an overview of tissue acidosis as it occurs in ischemic stroke, particularly in the ischemic penumbra. First, a concise mechanistic insight into ischemic tissue acidosis and acidotoxicity will be provided. Next, the utilization of tissue acidosis as a clinical tool to outline the ischemic penumbra will be presented. Finally, tissue acidosis as a guide to therapeutic, nanomedical intervention with the purpose to save the ischemic penumbra will be contemplated (Figure 1).

## 2. Mechanistic Insight into Ischemic Tissue Acidosis and Acidotoxicity

### 2.1. Mechanisms to Cause Cerebral Ischemic Tissue Acidosis

The shortage of metabolic substrates and oxygen, a condition that characterizes cerebral ischemia leads to metabolic acidosis. The limited availability of oxygen favors anaerobic glycolysis: pyruvate is reduced to lactate at the concomitant generation of a proton, which causes lactic acidosis [31]. In turn, tissue pH after cerebral ischemia onset decreases following an inversely linear relationship with tissue lactate concentration [31,32,33,34]. In addition, tissue pCO_2_ rises up to 3–4 fold, which may also contribute to tissue acidosis [35].

Even though astrocytes were initially thought to be a major source of acid production under cerebral ischemia [36,37], this interpretation was later contended and compartmentation of H^+^ was found to be negligible in the ischemic nervous tissue [34,38]. Intra- and extracellular acidosis with SD acquired at tissue level also displayed corresponding kinetics [19], indicating that acidosis in the intra-and extracellular space must be approximately matching in the metabolically challenged tissue. Indeed, intra-and extracellular pH in the ischemic nervous tissue probably equilibrates rapidly, because lactate newly produced in neurons and astrocytes is quickly released into the extracellular space as lactic acid via lactate/proton cotransport [39]. Metabolic acidosis may therefore not be restricted to specific cellular compartments.

The spatial and temporal kinetics of metabolic acidosis changes dynamically with the progression of ischemia. As such, tissue pH in the ischemic core may become as low as pH 6.0, while tissue pH fluctuates around pH 6.5–6.9 in the peri-infarct penumbra, as estimated in the acute middle cerebral artery occlusion (MCAO) rodent model of focal ischemic stroke [40,41]. In case the tissue is reperfused, tissue pH in the ischemic penumbra may display an alkalotic shift (pH 7.63) in the first hour post-ischemia, which is followed by gradual acidosis over the following hours (down to pH 6.58), coincident with infarct maturation [41,42]. In most studies dedicated to the exploration of cerebral ischemia-related tissue acidosis, the occurrence and contribution of SD has been overlooked [35,40,43]. It is important to recognize that the occurrence of SD transiently aggravates tissue acidosis, because SD propagating over the ischemic penumbra causes lactate load additional to that associated with hypoperfusion alone [18,44,45]. SD thus causes a decrease of tissue pH by around 0.3–0.5 pH units, taking off from mild or moderate acidosis [18,19] (Figure 2). For example, pH 6.9–7.1 was measured in the rat parietal cortex shortly after the bilateral occlusion of the common carotid arteries, but prior to the spontaneous occurrence of SD [19]. The SD emerging within minutes after ischemia onset aggravated tissue acidosis and decreased tissue pH down to 6.5 in average [19]. Further, the duration of the SD-related acid burden corresponds to SD duration lasting for a few minutes under ischemic penumbra conditions [46], which is increasingly longer in tissue zones undergoing more severe metabolic crisis [14,17]. Moreover, the SD-related acid load is remarkably extended with aging in the ischemic nervous tissue [46], which may contribute to the age-related acceleration of ischemic lesion maturation [47]. Finally, recurrent SDs in close succession may pose a sustained lactic acid load [20,48]. This is of importance because the prolongation of acid exposure is understood to lower the threshold of acidosis induced cell death [49]. Overall, SD may be perceived as a central mechanism of ischemic tissue acidosis.

### 2.2. Mechanistic Insight to Acidosis-Linked Neuronal Injury in Cerebral Ischemia

Excessive acidosis in the brain has long been considered to cause neuronal injury [32,49]. Furthermore, the detrimental impact of ischemic tissue acidosis on astrocyte function and survival has also been extensively studied [50]. The traditional view holds that acute acidosis in the nervous tissue leads to free radical generation, the failure of mitochondrial respiration, impaired protein synthesis, compromised cellular Ca^2+^ buffering, the disruption of intracellular signal transduction pathways, and the induction of DNA fragmentation [32,51,52,53,54]. In addition, tissue acidosis in cell culture has been shown to activate cytokine receptors and inflammatory pathways implicated in delayed neuronal injury subsequent to hypoxia [55].

The most recent mechanistic understanding of acidosis-linked neuronal death centers on the role of specific pH sensitive ion channels and exchangers, and culminates in intracellular Ca^2+^ accumulation or cytotoxic edema. As such, acid-sensing ion channel 1a (ASIC1a), a non-selective, proton-gated, amiloride-sensitive cation channel is activated by the reduction of extracellular pH [56,57] (Figure 3). Neuronal ASIC1a was shown to give way to a non-voltage-gated influx of Ca^2+^ in response to acidosis, which was suggested to contribute to the intracellular Ca^2+^ overload to initiate ischemic cell death [58,59,60]. Next, NMDA receptor signaling was found to enhance the ASIC1a-mediated Ca^2+^ current under ischemia, because Ca^2+^/calmodulin-dependent protein kinase II phosphorylated ASIC1a as a consequence of NMDA receptor activation [61]. Finally, ASIC1a was implicated in programmed necrosis via the recruitment of the cell death regulator serine/threonine kinase receptor interaction protein 1 (RIPK1) to ASIC1a in response to acidosis [62]. Importantly, ASIC1a inhibition proved to be neuroprotective in the MCAO rodent model of focal cerebral ischemia [58], providing further support to its implication in ischemic neuronal injury.

In addition to ASIC1a, a novel proton-activated Cl^−^ (PAC) channel has recently been attributed an acidotoxic role in cerebral ischemia [63] (Figure 3). The pathogenic potential of PAC channels lies in that the influx of Cl^−^ in response to acidosis causes cell swelling, and subsequent oncotic cell death [64]. The genetic knock-out of PAC channels provided partial neuroprotection in cultured neurons exposed to acidosis, and in the MCAO focal cerebral ischemia model [63,65]. Of note, the persistent and detrimental swelling of astrocytes exposed to lactacidosis was also linked to Cl^−^ [66], in part because the volume regulating Cl^-^ efflux through normally operational volume-sensitive Cl^-^ channels seemed to be inhibited in the combined presence of lactate and low pH [67].

While ASIC1a and PAC channels are thus activated by extracellular acidosis, intracellular acidification, as it occurs in cerebral ischemia, stimulates the operation of the Na^+^/H^+^ exchanger isoform 1 (NHE1), which is abundant on neurons [57] (Figure 3). NHE1 function is augmented by the ischemia-related phosphorylation of the exchanger, through the extracellular signal-regulated kinase (ERK)/90-kDa ribosomal S6 kinase (p90RSK) signaling pathway [68]. As a result, Na^+^ enters the cells in return for H^+^ extrusion, which attracts water to cause cell swelling [69]. Perhaps more importantly, the intracellular accumulation of Na^+^ reverses the Na^+^/Ca^2+^ exchanger to expel surplus Na^+^, and increases concomitantly the intracellular concentration of Ca^2+^ [70]. This then promotes cell death cascades linked to intracellular Ca^2+^ accumulation [57]. As with ASIC1a, the genetic ablation or direct pharmacological inhibition of NHE1 decreased infarct volume, reduced apoptosis, or limited Zn^2+^ accumulation-related neurodegeneration in experimental focal or global cerebral ischemia [70,71,72].

As a conclusion of these investigations, the blockade of ASIC1a, PAC channels or NHE1 has been proposed as a therapeutic opportunity in ischemic neuroprotection, with the aim to limit acidosis-linked neuronal injury [42,58,63,73].

## 3. Cerebral Tissue Acidosis in Theranostics and Nanomedicine

Brain imaging (computed tomography—CT, and magnetic resonance imaging—MRI) is a first step in the diagnosis of ischemic stroke. Beyond the primary assessment of the nature (i.e., ischemic or hemorrhagic), location, and volume of the injury of cerebrovascular origin, MRI and positron emission tomography (PET) techniques have been central to identify the ischemic penumbra, prognosticate its evolution, aid personalized therapeutic decision making, and confirm the fate of the penumbra tissue after treatment [6,74]. The reliable differentiation of the ischemic penumbra from the infarction or benign oligemic regions is, therefore, critical, and establishes the need to invent and refine applicable imaging tools [5]. The most recent developments in this field suggest that tissue acidosis can be used to distinguish penumbra tissue from the ischemic core with confidence, and the inclusion of pH imaging among imaging modalities used in stroke diagnostics has been recommended to bring informed decisions on patient care [6].

Obviously, the ischemic penumbra forms the central target of stroke therapy. Although recanalization is clearly intended to reperfuse and ideally save the ischemic penumbra, the delivery of pharmacological agents selectively to the penumbra zone is problematic, and remains a field for exploration. Next to narrow therapeutic time-windows, obvious difficulties hamper drug delivery to the ischemic territory, including the vascular occlusion blocking the direct vascular route of drug delivery, and the selective permeability of the blood-brain barrier (BBB) if the BBB remains intact. Yet, the residual blood flow typical of the ischemic penumbra may be sufficient in case the efficacy of drug delivery is amplified with specific drug carrier and drug release systems that (i) can cross the intact BBB, (ii) respond to the ischemic environment, and (iii) increase the local concentration of the therapeutic agent. Intriguingly, cancer therapy has already identified low pH typical of the tumor environment to direct anticancer drug delivery selectively to a tumor to enrich the tumor tissue with anticancer agents [75,76]. An analogous approach is thought to open up new possibilities in ischemic stroke therapy [77,78], and may advance the management of ischemic stroke in the future.

### 3.1. Tissue Acidosis to Identify Regions at Risk of Ischemic Damage

Fluorescent pH indicators have long been employed to visualize the spatial distribution of ischemic tissue acidosis in experimental models of cerebral ischemia. The important first studies to present pH images of coronal rodent brain slices used the pH sensitive fluorophores umbelliferone or neutral red, with the limitation of providing no temporal resolution of pH variation because the brains were frozen in situ at a chosen point of time for the subsequent ex vivo visualization of the pH signal [35,79,80] (Figure 4). Later, live optical imaging of the neutral red-loaded cerebrocortical surface of anesthetized rodents allowed high spatio-temporal resolution of the fluorescent pH signal [81], to follow the evolution of ischemia-or SD-related acidosis [19,82,83] (Figure 4). Yet, inherent to the optical approach, deeper brain regions escaped visualization, and the procedure of creating a craniotomy to serve imaging is highly invasive.

Much of the limitations of fluorescent indicator-based optical imaging appeared to be resolved with the utilization of PET using tracers for pH measurement [84,85,86], and amide proton transfer (APT) MRI, a type of chemical exchange saturation transfer imaging to visualize tissue acidosis [87,88]. Importantly, these imaging modalities focusing on acidosis have been applicable for stroke patients. Tissue pH imaging with APT-MRI have been found to define the salvageable ischemic penumbra with more confidence and better accuracy than the mismatch between perfusion- and diffusion-weighted imaging, which have been recognized to delineate the outer and inner edge of the ischemic penumbra, respectively [6,89]. Furthermore, APT-MRI was proposed as an imaging biomarker of clinical stroke symptoms and treatment efficacy [90]. Finally, pH sensitive nanoparticles used as image contrast enhancers to aid accurate distinction between diseased and normal tissue have been promising diagnostic tools under development. For example, PET imaging that used pH sensitive ^64^Cu-labelled polymers successfully delineated small tumors characterized by acidic pH in the mouse brain [91]. Also, Fe_3_O_4_ nanoparticles were demonstrated to be delivered to acidic ischemic brain tissue with pH responsive polymeric micelles to enhance MR images in the rat MCAO model [92]. Taken together, metabolic acidosis as it occurs in ischemic stroke emerges as a sensitive metabolic indicator to be potentially used for the identification of the penumbra tissue at risk of being recruited to the irreversibly damaged infarction, and to enable the design of personalized intervention in stroke therapy.

### 3.2. Tissue Acidosis to Guide Neuroprotective Intervention in Ischemic Stroke

The application of biocompatible and biodegradable, natural or synthetic macromolecular polymeric nanocarriers offers substantial promise in therapeutics [93,94,95]. Among others, stimulus responsive nanoparticles present the opportunity to initiate drug release by local (patho)physiological biochemical stimuli (e.g., homeostatic, redox, enzymatic, tissue pH) [96,97,98], which are intrinsic and restricted to the diseased tissue, and follow the progression of the disease condition. These bioresponsive nanomaterials are also known as “smart” nanosystems [29]. A negative pH shift from the neutral 7.3–7.4 to below 7.0 units, for instance, can initiate conformational or solubility changes in various smart nanosystems, including polysaccharide chitosan nanoparticles, to allow drug release [29,76,78]. In accordance, the acidic local tumor environment created by intensive or dysregulated glucose metabolism [99,100,101] has been utilized as a specific trigger for drug release in the treatment of solid cancers [76,102,103].

The achievements of cancer nanomedicine have inspired the application of nanotechnology in the therapy of ischemic stroke, especially because these diverse disease entities share some distinct pathophysiological processes [29]. Along with the disintegration of microvascular ultrastructure, intensified generation or failing clearance of reactive oxygen species, and cellular immune reactions [29], tissue acidosis occurs in tumors [99,100,101], as well as in ischemic brain tissue. Building on our experience gained in the field of metabolic tissue acidosis associated with SD in the ischemic cerebral cortex [18,71], we designed and constructed pH-responsive chitosan nanoparticles [78]. After it had been confirmed in suspension that the nanoparticles released nimodipine when pH fell below pH 7.0 [78], we tested the efficacy of drug delivery with the nanoparticles in a preclinical model of cerebral ischemia [77] (Figure 5). Our working hypothesis posited that drug carrier, pH-sensitive nanoparticles should release their cargo in response to acidic tissue pH (<pH 7.0) typical of the ischemic penumbra or SD-affected nervous tissue. We have selected nimodipine, an L-type voltage-gated Ca^2+^ channel antagonist as the drug to be delivered, because the cerebral vasodilator, SD limiting and ischemic neuroprotective actions of nimodipine have been widely acknowledged [104,105,106,107,108], and reproduced in our lab [109,110], to be used as a reference for the nanoparticle study. In the experiments, the typical cerebral vasodilator effect of nimodipine carried by the nanoparticles became apparent only once tissue acidosis with ischemia onset (drop of pH from about 7.29 to 6.9–7.1) or SD (tissue pH down to 6.7) had occurred. Further, nimodipine suppressed SD and augmented the related CBF response [77], as expected [110] (Figure 5). On the basis of these results, the important principle was formulated that tissue acidosis, as it occurs in cerebral ischemia and in association with SD, could be used as a trigger for drug delivery by a smart, bioresponsive nanosystem.

These results are encouraging, and also raise a number of further considerations for the potential biomedical application of the principle of acidosis guided drug targeting. We washed the nanoparticle suspension directly to the exposed cortical surface in our pre-clinical model, which offers data relevant for potential intracerebroventricular, intraparenchymal or intrathecal drug delivery [27]. Yet, all these methods are significantly invasive and carry the risk of inflammation. Therefore other routes of administration that are more realistic in routine clinical care need to be tested. An obvious option appears to be intravenous infusion. For this approach, the BBB permeation of the chitosan nanospheres must be evaluated, because particles larger than 12–30 nm may not cross the BBB [111]. Further, the retention of nanoparticles in non-target tissues (e.g., cells of the reticuloendothelial system) could decrease the amount of circulating nanoparticles before their penetration to the brain [112]. Finally, potentially low tissue pH prevailing in peripheral organs or body fluids (e.g., in the respiratory system or the gastrointestinal tract) would perceivably cause off-target drug release. Although the size of the nimodipine-loaded nanoparticles in our study was small enough for BBB penetration (i.e., 4–6 nm) [78], the BBB permeability of chitosan nanoparticles may be improved by functionalizing chitosan with antibodies that recognize receptors specific to BBB endothelial cells (e.g., transferrin receptors) [113,114]. This should initiate the receptor-mediated transcytosis of the nanospheres. In addition, cerebral ischemia may derange the BBB and enhance non-selective transendothelial vesicular transport, or loosen the tight junctions between adjacent endothelial cells [115]. This may allow drug carriers to reach the nervous tissue along with the extravasation of blood plasma. Of note, SD itself is capable of increasing endothelial transcytosis and paracellular diffusion at the BBB [116,117,118], and was found to facilitate drug delivery to the brain tissue [117]. Finally, the intracarotid, rather than intravenous route of infusion of drug-loaded nanocarriers should provide direct access to the brain [119], which could substantially reduce off-target retention and drug release at the periphery.

The intranasal application of the nanoparticles may be an alternative route of drug administration. Chitosan, in fact, displays very good adhesion to the nasal mucosa due to the positively charged nanoparticle surfaces [120], and enhances absorption through the nasal mucosa by disrupting the intercellular tight junctions of the epithelium [121,122]. However, the nasal mucosa is acidic (pH 5.5–6.5). This condition counter-indicates the nasal administration of acid responsive nanoparticles targeting the brain, unless the nanoparticles are applied in a pH neutralizing buffer medium, or are supplied with a protective coating during their passage through the nasal mucosa, which the particles should shed before reaching the brain.

In summary, ischemic tissue acidosis in the brain may be utilized to direct bioresponsive nanocarriers to release their drug cargo in response to acidic tissue pH. This approach would ensure that pharmacological agents are targeted selectively to tissue at risk of injury in ischemic stroke, with minimizing off-target effects. In fact, ASIC1a, PAC channel or NHE1 blockers that counteract acidotoxicity (see under Mechanistic insight to acidosis-linked neuronal injury in cerebral ischemia) may be delivered driven by tissue acidosis itself.

## 4. Conclusions

In summary, a key mechanism of tissue acidosis as it occurs in focal cerebral ischemia is the accumulation of lactic acid, which is largely achieved by SD. Tissue acidosis is a sensitive metabolic indicator of injury progression in cerebral ischemia, which makes acidosis suitable for the identification of the salvageable ischemic penumbra. Further, acidosis exacerbates ischemic injury in the nervous tissue. The acidotoxic cellular Ca^2+^ accumulation and cytotoxic edema, which are mediated by ion channels sensitive to an acidic shift in tissue pH (e.g., ASIC1a, PAC channels, NHE1), may be limited with the use of selective ion channel blockers. These agents with neuroprotective promise are suggested here to be delivered selectively to the ischemic penumbra with pH-responsive smart nanosystems. Tissue pH-targeted drug delivery is expected to enrich sites of ongoing injury with the therapeutical agent, without the risk of unfavorable off-target side effects.

## Figures and Tables

**Figure 1 biology-09-00460-f001:**
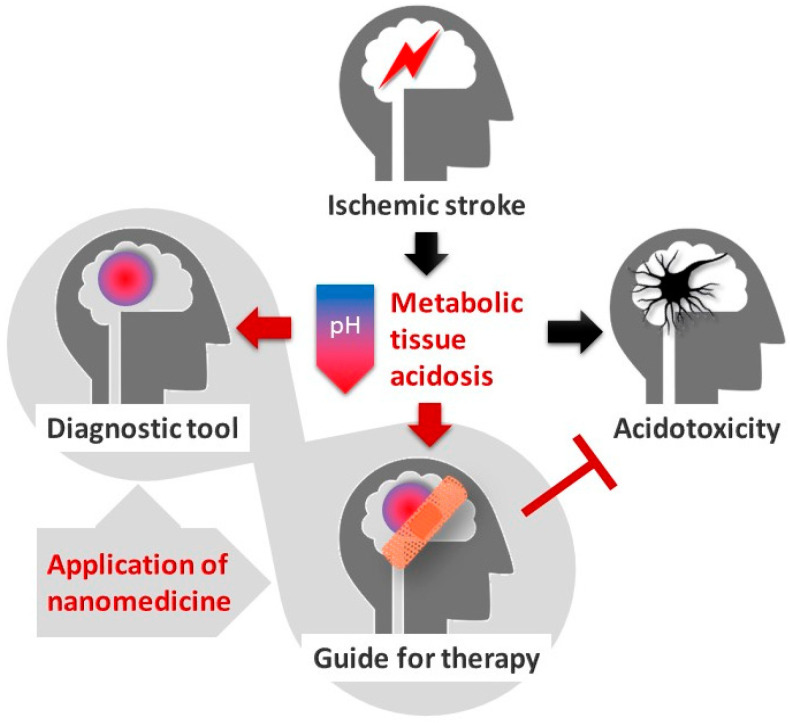
The stroke-associated metabolic acidosis of the nervous tissue, which is lethal to neurons above a critical threshold (“Acidotoxicity”), may be harnessed for the accurate identification of the ischemic penumbra (“Diagnostic tool”), or for focusing drug delivery selectively to the ischemic penumbra (“Guide for therapy”) with smart nanosystems.

**Figure 2 biology-09-00460-f002:**
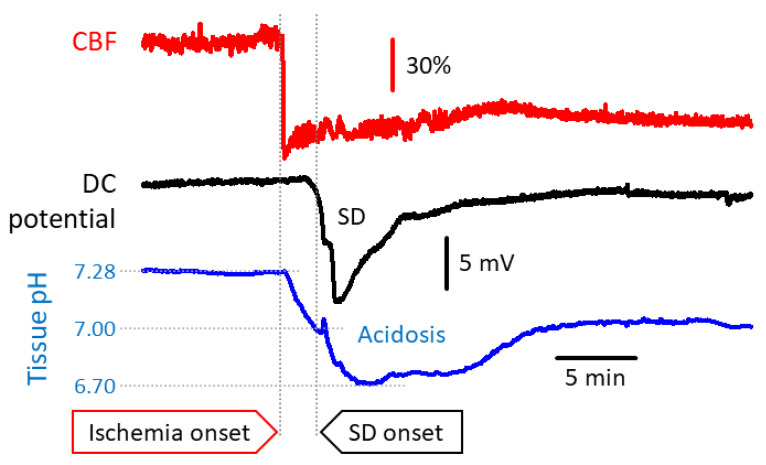
Representative traces demonstrate the association of tissue acidosis (**blue**) with ischemia onset (bilateral common carotid artery occlusion; (**red**) and spreading depolarization (SD); (**black**) in the rat parietal cortex. Tissue pH was measured with a pH-sensitive microelectrode implanted in the cortex, cerebral blood flow (CBF) was monitored with laser Doppler flowmetry, and the DC potential was acquired with an intracortical glass capillary microelectrode.

**Figure 3 biology-09-00460-f003:**
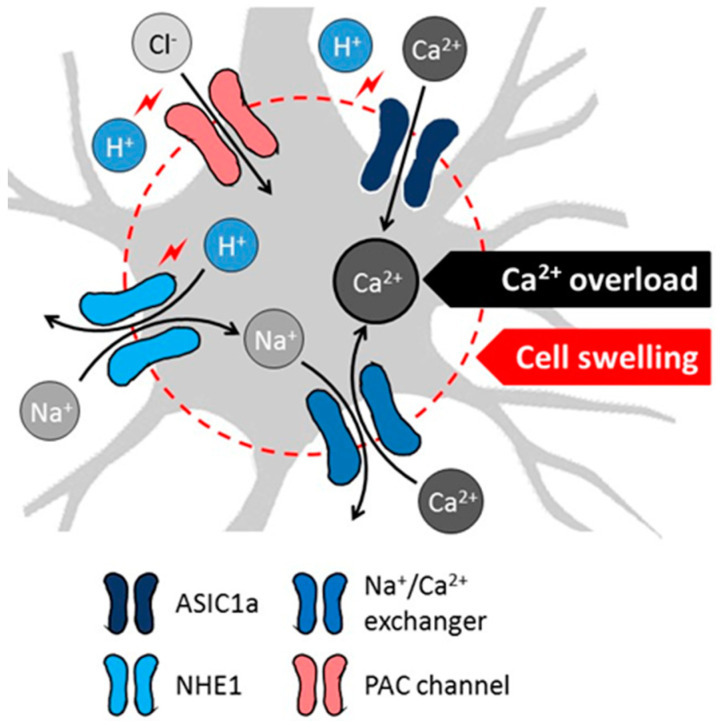
Proposed cellular mechanisms of acidotoxicity in ischemic stroke include acidosis-related intracellular Ca^2+^ overload and swelling of neurons (schematic gray-shaded cell). Decreasing extracellular pH activates the acid-sensing ion channel 1a (ASIC1a) to allow direct Ca^2+^ influx to neurons. Low intracellular pH facilitates H^+^/Na^+^ exchange, followed by the elimination of surplus intracellular Na^+^ via Na^+^/Ca^2+^ exchange. The ion movements lead to the ultimate intracellular accumulation of Ca^2+^. Low extracellular pH also initiates Cl^−^ influx through proton-activated Cl^−^ (PAC) channels. The movement of Na^+^ (via NHE1) and Cl^−^ (via PAC channels) into the neurons attract water, increase the water content of cells and cause cytotoxic edema. The mechanisms appear to be relevant for the SD-related tissue acidosis.

**Figure 4 biology-09-00460-f004:**
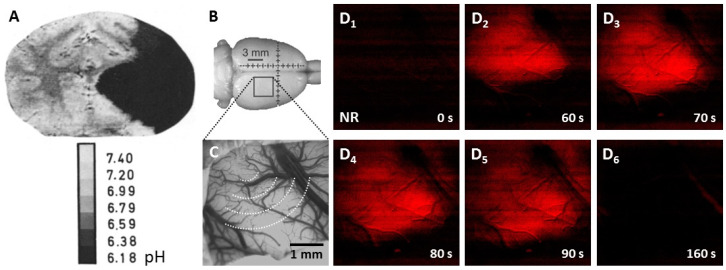
Experimental tissue pH imaging in models of cerebral ischemia and spreading depolarization. (**A**), Umbilliferone fluorescence pH imaging 2 h after experimental middle cerebral artery occlusion in the cat. Reprinted from Csiba et al., 1983 [80], with permission from Elsevier. (**B**–**D**), Neutral red fluorescence imaging of spreading depolarization (SD) through a closed cranial window preparation over the parietal cortex of an anesthetized rat. (**B**), The position of the closed cranial window. (**C**), An intrinsic optical signal image of the exposed cortical surface at green light illumination. The schematic radial hemi-circles indicate the wave of SD. (**D**), Transient tissue acidosis propagating with SD, depicted by the increasing intensity of the Neutral red fluorescent signal (red) in background-subtracted, contrasted and pseudo-colored images. Time with respect to SD initiation is shown in the lower right corner of the images.

**Figure 5 biology-09-00460-f005:**
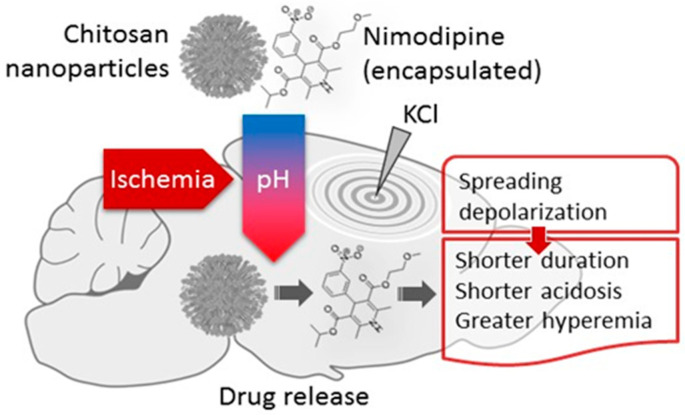
Nimodipine delivered with pH-responsive chitosan nanoparticles to the cerebral cortex suppressed spreading depolarization (SD) and augmented the hyeremic element of the associated CBF response in a rat incomplete global forebrain ischemia model. Reprinted from Tóth, M. et al., 2020 [77], with permission from Elsevier.

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
