# Peer review of "Tissue Acidosis Associated with Ischemic Stroke to Guide Neuroprotective Drug Delivery"

_biology, 2020, doi:10.3390/biology9120460_

Round 1
Reviewer 1 Report
Here Orsolya et al. discuss an attractive therapeutic approach against acute ischemic stroke. Specifically, the review includes a general introduction of ischemic stroke and tissue acidosis triggered by this injure. Next, the authors propose to target the penumbra area with “smart” nanomedicine that releases the therapeutic agent in the presence of an acid environment. Finally, Orsolya et al. summarize their previously published approach, nimodipine encapsulated in chitosan nanoparticles, and propose strategies to improve the delivery to the brain.
Please see bellow some minors observations:
1 Line 35 should include the reference
2 In figure 3, what does it mean the lighting icon?
3 Authors should describe or define hyperemia (figure 4)
Author Response
Response to the Reviewer’s suggestions – Reviewer #1
Manuscript ID: biology-1009353
Title: Tissue acidosis associated with ischemic stroke to guide neuroprotective drug delivery
Corresponding Author: Dr. Eszter Farkas
Authors: Orsolya M. Tóth, Ákos Menyhárt, Rita Frank, Dóra Hantosi, Eszter Farkas, Ferenc Bari
The encouraging feedback and the constructive comments of Reviewer #1 are much appreciated, and have been taken into account on preparing the revised version of the manuscript. In our best effort to comply with the Reviewer’s queries, we implemented a careful revision of the manuscript.
The changes to the manuscript within the document have been highlighted by using colored text (red font). In order to expedite the processing of the revised manuscript, we provide point-by-point responses to the Reviewer’s comments as follows.
Q1: Line 35 should include the reference
Authors’ response and changes in the manuscript: The required reference has been included.
Q2: In figure 3, what does it mean the lighting icon?
Authors’ response and changes in the manuscript: The gray-shaded background indicates a neuron. This has been described in the figure legend in the revised manuscript.
Q3: Authors should describe or define hyperemia (figure 4)
Authors’ response and changes in the manuscript: Spreading depolarization is associated with a profound cerebral blood flow response, in which a functional or reactive hyperemia is a defining element. This information has been provided in the figure legend of the revised manuscript.
We thank the Reviewer for the supportive Review Report!

Reviewer 2 Report
The article "Tissue acidosis associated with ischemic stroke to guide neuroprotective drug delivery" makes a very good summary on the understanding of post ischemic tissue acidosis and how to tackle it in order to apply novel therapeutic approaches in the always frustrating field of neuroprotection.
The field of stroke research and therapeutics remains severely hampered by the complexities that surround the protection of the tissue in the initial phases of stroke, therefore it is mandatory to develop new ideas like the ones described in this paper, that will provide new work frames.
Overall the paper is excellently written, the language is perfectly used, the topics are correctly introduced and well interconnected, the conclusions are accurate and summarize the contents very well.
Main question to the authors:
In reading your paper I realized the topic is (correctly) very well focused on tackling the stroke biology in the penumbra and during the subacute phase of stroke. Nevertheless, a large part of neuronal death also takes part during the recovery. One can observe mice waking up from their surgery showing relatively minor effects (except for one sided turning), only to show a severe worsening in behavior over the next weeks which then turns into a mild recovery over the next 3 months, which can be potentiated by long term therapeutic interventions. During this phase the critical point is to promote neuroplasticity while at the same time reverting brain inflammation.
Would you speculate a little bit on how tissue acidosis may be controlled in order to promote neuronal plasticity during this period? I think that it would help to add a sentence or two during the introduction paragraphs to explain what happens with tissue acidosis after the subacute phase of the stroke and how it projects into recovery.
Minor comments:
- Abstract. Shorten or remove paragraph 2 and make a better description of the smart nano system, since it is the main attractor to the paper. The biology of tissue acidosis is deeply discussed in the paper itself and is also of common knowledge.
- Shorten the first three paragraphs of the introduction. The discussion about metabolic acidosis makes sense, but all the comments about failure of neuroprotective therapies, etc, are widely known facts that shorten the space for more significant discussions on the review´s topic.
- Remove quotes from figure legends. Not critical, but it's not customary.
- Line 99: “... in THE metabolically challenged tissue”.
- Line 141: Better would be “Neuronal ASIC1a…” Modify accordingly if you find other examples in the text.
- While ASIC1a and PAC channels are thus activated by extracellular acidosis, intracellular acidification, as it occurs in cerebral ischemia, stimulates the operation of the Na+/H+ exchanger 159 isoform 1 (NHE1), which is abundant on neurons [59] (please note commas and italic).
- Maybe add an image of the use of brain imaging and the use of acidosis to delineate the infarct area next to the graph depicting the mechanisms of tissue acidosis. If not from humans at least from mice tissue.
This is a very good paper, I hope it is published soon.
Author Response
Response to the Reviewer’s suggestions – Reviewer #2
Manuscript ID: biology-1009353
Title: Tissue acidosis associated with ischemic stroke to guide neuroprotective drug delivery
Corresponding Author: Dr. Eszter Farkas
Authors: Orsolya M. Tóth, Ákos Menyhárt, Rita Frank, Dóra Hantosi, Eszter Farkas, Ferenc Bari
The encouraging feedback and the constructive comments of Reviewer #2 are much appreciated, and have been taken into account on preparing the revised version of the manuscript. In our best effort to comply with the Reviewer’s queries, we implemented a careful revision of the manuscript.
The changes to the manuscript within the document have been highlighted by using colored text (red font). In order to expedite the processing of the revised manuscript, we provide point-by-point responses to the Reviewer’s comments as follows.
Q1: (…) Would you speculate a little bit on how tissue acidosis may be controlled in order to promote neuronal plasticity during this period? I think that it would help to add a sentence or two during the introduction paragraphs to explain what happens with tissue acidosis after the subacute phase of the stroke and how it projects into recovery.
Authors’ response and changes in the manuscript: The suggestion is much appreciated. The temporal aspects of ischemic issue acidosis and the linked, delayed neuronal injury has been contemplated at two sections in the manuscript.
In “2.2 Mechanistic insight to acidosis-linked neuronal injury in cerebral ischemia”, the addition runs as:
“In addition, tissue acidosis in cell culture has been shown to activate cytokine receptors and inflammatory pathways implicated in delayed neuronal injury subsequent to hypoxia (Frøyland et al., 2008).”
In “2.1 Mechanisms to cause cerebral ischemic tissue acidosis”, the addition runs as:
“In case the tissue is reperfused, tissue pH in the ischemic penumbra may display an alkalotic shift (pH 7.63) in the first hour post-ischemia, which is followed by gradual acidosis over the following hours (down to pH 6.58), coincident with infarct maturation (Back et al., 2000; Pinataro et al., 2007).”
Q2: Abstract. Shorten or remove paragraph 2 and make a better description of the smart nano system, since it is the main attractor to the paper. The biology of tissue acidosis is deeply discussed in the paper itself and is also of common knowledge.
Authors’ response and changes in the manuscript: The suggestion is welcome. Paragraph 2 of the Abstract has been shortened, and the space gained has been used to describe the pH-responsive drug delivery in more detail.
Q3: Shorten the first three paragraphs of the introduction. The discussion about metabolic acidosis makes sense, but all the comments about failure of neuroprotective therapies, etc, are widely known facts that shorten the space for more significant discussions on the review´s topic.
Authors’ response and changes in the manuscript: The suggestion is appreciated. The Introduction has been shortened to avoid space occupying, widely known facts.
Q4: Remove quotes from figure legends. Not critical, but it's not customary.
Authors’ response and changes in the manuscript: The quote from the legend of Figure 1 has been removed. The quote has been kept in the legend of Figure 5, to acknowledge the original source for copyright reasons.
Q5: Line 99: “... in THE metabolically challenged tissue”.
Authors’ response and changes in the manuscript: Thank you – the grammar has been corrected.
Q6: Line 141: Better would be “Neuronal ASIC1a…” Modify accordingly if you find other examples in the text.
Authors’ response and changes in the manuscript: The suggestion is appreciated, the modification has been implemented.
Q7: While ASIC1a and PAC channels are thus activated by extracellular acidosis, intracellular acidification, as it occurs in cerebral ischemia, stimulates the operation of the Na+/H+ exchanger 159 isoform 1 (NHE1), which is abundant on neurons [59] (please note commas and italic).
Authors’ response and changes in the manuscript: Thank you – the grammar has been corrected.
Q8: Maybe add an image of the use of brain imaging and the use of acidosis to delineate the infarct area next to the graph depicting the mechanisms of tissue acidosis. If not from humans at least from mice tissue.
Authors’ response and changes in the manuscript: The suggestion is welcome! An additional figure, now Figure 4 demonstrates umbilliferone and Neutral red fluorescence imaging of an experimental focal infarct and spreading depolarization, respectively.
We thank the Reviewer for the supportive Review Report!
